# Evolution of the Electronic Structure of the *trans*-[Re_6_S_8_bipy_4_Cl_2_] Octahedral Rhenium Cluster during Reduction

**DOI:** 10.3390/molecules28093658

**Published:** 2023-04-23

**Authors:** Maxim R. Ryzhikov, Yakov M. Gayfulin, Anton A. Ulantikov, Dmitry O. Arentov, Svetlana G. Kozlova, Yuri V. Mironov

**Affiliations:** Nikolaev Institute of Inorganic Chemistry SB RAS, 3, Acad. Lavrentiev Ave., 630090 Novosibirsk, Russia

**Keywords:** octahedral rhenium clusters, reduction, DFT, ELF, cyclic voltammetry

## Abstract

Understanding the processes that occur during the redox transformations of complexes coordinated by redox-active apical ligands is important for the design of electrochemically active compounds with functional properties. In this work, a detailed analysis of the interaction energy and electronic structure was performed for cluster complexes *trans*-[Re_6_S_8_bipy_4_Cl_2_]^n^ (n = 2–, 4–, 6–, 8–), which can be obtained by stepwise electrochemical reduction of a neutral cluster *trans*-[Re_6_S_8_bipy_4_Cl_2_] in DMSO solution. It was shown that the formation of open-shell paramagnetic ions with S = 1, 2 and 1 is the most energetically favorable for n = 2–, 4– and 6–, respectively.

## 1. Introduction

The design of electrochemically active complexes of transition metals has been a frontier area of inorganic chemistry in recent decades. Interest in the preparation of such complexes is caused by their potential for use as materials for electrocatalysis and photocatalysis, optical applications, and the preparation of redox-active coordination polymers and compounds with cooperative magnetic effects [1,2,3,4]. The study of soluble molecular redox-active compounds as electronic reservoirs and charge carriers in chemical current sources has also become an important area of research [5]. Existing approaches to the design of redox-active compounds in most cases include the preparation of mononuclear or polynuclear transition metal complexes with redox-active (non-innocent) ligands [6,7,8,9]. This strategy makes it possible to obtain complexes capable of multi stage oxidation and reduction in a wide range of applications. In these processes, many effects arise that are interesting for research from a fundamental point of view. Such effects, in particular, are the mutual influence of the metal cation and the ligand on the electronic structure and the spectroscopic properties and stability of the oxidized and reduced forms of the complexes.

In the past few years, a great amount of attention has been attracted to redox-active transition metal cluster compounds and, in particular, octahedral clusters of {M_6_Q_8_} type [10,11,12,13,14,15]. These complexes are based on an octahedral core consisting of six metal cations linked to each other by covalent bonds. The metal core is coordinated by a set of eight “inner” ligands lying on the faces of the octahedron, and six apical ligands, one for each metal atom. As a result, the general formula for these cluster complexes can be written as [{M_6_Q_8_}L_6_]^n^, where Q and L denote inner and apical ligands, respectively. The presence of covalent bonds in the cluster core leads to a unique electronic structure where the atomic orbitals of the metal atoms and inner ligands overlap and form a set of frontier orbitals, which are delocalized over all atoms of the cluster core [10]. Therefore, the cluster core can be considered a single metal ion coordinated by a set of apical ligands.

Cluster cores {M_6_Q_8_} are capable of oxidation, which is associated with a decrease in the number of electrons localized in bonding metal-centered molecular orbitals (cluster skeletal electrons, CSE). The effects of coordination of various N-donor [16,17,18,19], P-donor [20,21,22], and O-donor [23,24,25,26] apical ligands on the potentials of these oxidative transitions have been studied in detail. On the other hand, attempts to reduce the {M_6_Q_8_} cluster cores to 25 CSE led to irreversible processes associated with the destruction of the cluster core due to the strongly antibonding character of the lowest metal-centered unoccupied orbitals. However, as in the case of single metal ions, the coordination of redox-active (non-innocent) apical ligands to the cluster cores often makes it possible to obtain compounds capable of reversible reduction due to the presence of low-lying ligand-centered π* molecular orbitals. For example, {Mo_3_S_4_} clusters decorated with redox active diimine ligands demonstrate multistage reduction processes [27,28,29]. It was also shown that the coordination of pyridine derivatives to {Re_6_Q_8_} cluster cores often led to the obtaining of compounds exhibiting multielectron reduction processes in cyclic voltammograms [16,19]. All these clusters can act as multi electron reservoirs, which may allow them to be used in catalytic and magnetochemical applications.

In recent years, we have synthesized and investigated a series of new redox-active octahedral rhenium cluster complexes coordinated by redox-active N-donor ligands. Particularly, molecular clusters with the general formula *trans*-[Re_6_Q_8_L_4_X_2_] (Q = S or Se, X = Cl^−^, Br^−^ or CN^−^) were obtained for L = 4,4′-bipyridine, 4-phenylpyridine, *trans*-1,2-Bis(4-pyridyl)ethylene, and 1,3-Bis(4-pyridyl)propane [30,31,32,33]. Electrochemical investigation of the compounds *trans*-[Re_6_Q_8_bipy_4_X_2_] (Q = S or Se; X = Cl or Br; bipy = 4,4′-bipyridine) (Figure 1) showed that the complexes are capable of accepting up to eight electrons per ligand-centered molecular orbitals, each localized to two bipy molecules in the *trans*-position [30]. In this case, the chemical preparation of the reduced forms was difficult due to the low solubility of the compounds and the low potentials required. However, since the reduction of free 4,4′-bipyridine leads to the production of radical anions, the reduction of compounds *trans*-[Re_6_Q_8_bipy_4_X_2_] can potentially lead to the formation of paramagnetic radical anions with up to four unpaired electrons. The possibility of obtaining such polyradicals can stimulate further research in this area; therefore, the aim of this article was to determine the preferable electron configuration of [Re_6_S_8_bipy_4_Cl_2_]^n^ clusters, with n = 0, 2–, 4–, 6–, 8–, by DFT calculations and to trace changes in the electronic structure during reduction.

## 2. Computational Details

The geometry optimization and frequency calculations of the [Re_6_S_8_bipy_4_Cl_2_]^n^ clusters in n = 0, 2–, 4–, 6–, 8– oxidation states were performed in the ADF2017 [34,35] program suite with generalized gradient approximation (GGA) dispersion corrected S12g [36] density functional, all-electron STO’s TZP [37] basis set, and zero-order regular approximation (ZORA) [38] to take into account scalar relativistic effects. Since dimethyl sulfoxide (DMSO) was previously used for electrochemical investigations of the [Re_6_S_8_bipy_4_Cl_2_]^0^ cluster [30], it was used as the solvent for calculations. The conductor-like screening model (COSMO) [39] was used to take into account the DMSO environment. The spin-restricted (S = 0) approximation was used for the cluster in all oxidation states. The spin-unrestricted approximation was additionally used for 2– (S = 1), 4– (S = 1 and S = 2), and 6– (S = 1) oxidation states. Since the [Re_6_S_8_bipy_4_Cl_2_]^n^ clusters are characterized by eight pyridine heterocycles that could be difficult to optimize in internal coordinates, the Cartesian coordinate space was used for geometry optimization. The optimization process started with the C_i_ symmetry as it was found in the crystal structure of the neutral [Re_6_S_8_bipy_4_Cl_2_]^0^ cluster. However, some of the structures optimized at C_i_ symmetry have imaginary frequencies (typically with S = 0 spin state). In such a case, the symmetry was lowered to C_1_. Thus, it was possible to achieve structures without imaginary frequencies. Note that most of the structures optimized at C_1_ symmetry (n = 2–, S = 0; n = 6–, S = 0; n = 6–, S = 1 and n = 8–, S = 0) are still close to the C_i_ symmetry, with the root mean square (RMS) deviation from C_i_ not exceeding 0.04 Å. The only exception is the [Re_6_S_8_bipy_4_Cl_2_]^4–^ S = 0 cluster with a structure closer to C_2_ (still quite far RMS 0.07 Å). The geometry optimization calculations performed at the S12g/TZP level of theory typically gives good structural parameters for transition metal compounds in a reasonable amount of time [40,41,42,43]. The optimized coordinates of the [Re_6_S_8_Cl_2_bipy_4_]^n^ cluster in all oxidation and spin states are summarized in Appendix A.

Since GGA functionals typically underestimate the gap between occupied and unoccupied levels, single-point calculations with the S12h [35] dispersion-corrected hybrid density functional, all-electron TZ2P basis set, COSMO model for the DMSO environment, and scalar relativistic ZORA were performed on optimized geometries. The calculation and analysis of the electron localization function (ELF) [44,45] were performed in the dgrid-4.6 [46] program with a 0.05 a.u. mesh step. The atomic charges were calculated by the definition of the quantum theory of atoms in molecules (QTAIM) [47]. In the QTAIM, the atomic basin is defined as all points of the space in which the gradient line finishes in the atomic attractor (the local maximum of electron density located at the position of the atom nucleus). To obtain the atomic charges within QTAIM, the electron density must be integrated over the volume of the corresponding basin. The QTAIM charges were calculated with the built-in ADF tools with default settings. Since the integration is performed numerically over real space, an error could arise for heavy atoms, which have a strong nonlinearity of the electron density near the nucleus. The deviations of the total number of electrons after integration from the necessary value do not exceed 0.0008 e, indicating good integration grid accuracy.

The energy decomposition analysis (EDA) [48] calculations were performed in ADF2020 [49] to analyze the interaction energy (E_int_) between Re_6_S_8_Cl_2_ and [bipy_4_]^n^ fragments at the same theoretical level as single-point calculations. In all EDA calculations, the Re_6_S_8_Cl_2_ cluster fragment was taken as neutral and spin-restricted, while the charge and the spin state of the [bipy_4_]^n^ fragment were taken in accordance with the respective properties of the whole cluster [Re_6_S_8_Cl_2_bipy_4_]^n^.

## 3. Results and Discussion

It was shown that the reduction of the [Re_6_S_8_bipy_4_Cl_2_]^n^ cluster occurs in four waves in cyclic voltammetry (CV) experiments, and each wave corresponds to the transfer of approximately two electrons [30]. Therefore, the cluster was calculated with n = 0, 2–, 4–, 6– and 8–. The unoccupied frontier molecular orbitals (Figure 2) of the neutral [Re_6_S_8_bipy_4_Cl_2_]^0^ cluster are almost degenerate (E_LUMO+1 − LUMO_ = 0.021 eV and E_LUMO+3 − LUMO+2_ = 0.005 eV at S12h/TZ2P//S12g/TZP level of theory). Thus, the reduced states may have an open-shell electronic structure. The comparison of the relative energies (Table 1) shows that the most stable spin states are 1, 2, and 1 for the cluster in 2–, 4–, and 6– oxidation states, respectively. The cluster in (n = 0; S = 0), (n = 2–; S = 1), (n = 4–; S = 2), (n = 6–; S = 1), and (n = 8–; S = 0) oxidation and spin states will be used for subsequent study.

Four redox transitions in CV allow the comparison of the experimental half-wave potential (E½) values [30] with relative energies (ΔE) of the cluster in different oxidation states (Figure 3). The experimental and calculated points can be fitted by a straight line with 0.993 and 0.995 R^2^ values for S12g/TZP and S12h/TZ2P//S12g/TZP levels of theory, respectively. Such R^2^ values indicate that the electronic structure changes occurring in the cluster during electrochemical reduction can be reproduced with a good accuracy at both theoretical levels.

The formation of the four lowest unoccupied molecular orbitals of the neutral [Re_6_S_8_bipy_4_Cl_2_]^0^ cluster can be traced starting from the LUMO orbital of individual bipy ligands, through the model fragment of four bipy ligands to the complete cluster (Figure 4). As can be seen, the LUMO of the two bipy molecules in *trans*-positions solely forms the LUMO, LUMO + 1, LUMO + 2, and LUMO + 3 of the bipy_4_ fragment. These MOs are composed of bipy’s LUMO with an almost equal contribution, making them very close in energy (ΔE = 0.01 eV). It is clear that the pairing of bipy’s LUMO in *trans*-positions is explained by the number of possible combinations. Four LUMOs of bipy in *trans*-positions can form the four MOs (two bonding and two antibonding) (Figure 4), while four LUMOs of bipy in *cis*-positions can form eight MOs (four bonding and four antibonding) (Appendix A), which did not correlate with the number of original states (four LUMO of bipy). The addition of the Re_6_S_8_Cl_2_ cluster fragment to the bipy_4_ fragment slightly changes the order of the four lowest unoccupied MOs. However, the four lowest unoccupied orbitals of the [Re_6_S_8_bipy_4_Cl_2_]^0^ cluster are still composed mainly of the orbitals of the bipy_4_ fragment.

The interatomic distances in the cluster core (d(Re–Re) and d(Re–S)) of the [Re_6_S_8_bipy_4_Cl_2_]^n^ clusters are almost constant during the reduction process (Table 1). The distances between rhenium atoms and the atoms of the terminal ligands are more sensitive to the reduction. The unoccupied orbitals of the neutral [Re_6_S_8_bipy_4_Cl_2_]^0^ cluster, which is populated during reduction (LUMO, LUMO + 1, LUMO + 2 and LUMO + 3), are primary localized on the bipy ligands. However, the Re–Cl distances become 0.9 Å longer for eight-electron reduction, indicating significant bond weakening (Table 1). In contrast, the Re–N_in_ bonds display a large shortening from 2.21 Å to 2.12 Å during reduction of the cluster, which correlates with the contribution of the bonding Re–N_in_ interaction in the four lowest unoccupied MOs (Figure 4).

Based on the calculated energies (Table 1) and the molecular orbitals (Figure 5) of the cluster with the different charges, the following process of reduction can be proposed. The pair of molecular orbitals delocalized over bipy ligands in trans position are occupied by a single electron each during the first two-electron reduction wave, and the resulting [Re_6_S_8_bipy_4_Cl_2_]^2−^ cluster has two unpaired electrons. The second two-electron reduction wave leads to the occupation of the two additional MOs by a single electron each, with the formation of the [Re_6_S_8_bipy_4_Cl_2_]^4−^ cluster with four unpaired electrons. In this case, all four of the bipy_4_-centered orbitals become populated by a single electron each. During the consequent reduction wave, two orbitals already populated by the single electron are occupied by two additional electrons, reducing the number of unpaired electrons to two in the [Re_6_S_8_bipy_4_Cl_2_]^6−^ cluster. The final wave causes all four orbitals to become populated by two electrons each in the [Re_6_S_8_bipy_4_Cl_2_]^8−^ cluster. Thus, the four bipy_4_-centered orbital can roughly be considered as a separate electronic shell that populates according to Hund’s rule. Finally, the eight-electron reduction by the four two-electron reduction waves is shown in Figure 1.

Since the MOs in the [Re_6_S_8_bipy_4_Cl_2_]^0^ cluster are delocalized over a large number of atoms, changes in bonding during the reductions are difficult to trace by the MO analysis. ELF analysis was used to reveal changes in bonding, as it was shown to give good results for other cluster compounds [50]. The ELF basin pattern in the {Re_6_} cluster core is quite stable during the eight-electron reduction (Table 2). In all oxidation states, there are 12 disynaptic V(Re,Re) basins with an average population of 0.53–0.54 e and eight trisynaptic V(Re_3_) basins with an average population of 0.16 e. The V(Re,Re) and V(Re_3_) basins indicate the two-center and three-center covalent interactions, respectively. At some oxidation states, there are also polysynaptic V(Re_6_) and V(Re_5_) basins, but the populations of such basins are negligible. The most pronounced changes are found for V(Re,N_in_) basins (Appendix A), in which populations grow from 2.59 e in the neutral cluster to 3.45 e in the octa-charged anionic cluster. Note that the charges on the N_in_ and Re atoms did not change substantially (Appendix A). The population of V(N_in_,C) basins decreases from 2.42 to 2.01 e during reduction due to the antibonding nature of LUMO, LUMO+1, LUMO+2, and LUMO+3 relative to the N_in_–C interactions. Thus, four V(Re,N_in_) basins in total took ~3.5 additional electrons due to the redistribution of electron density from V(N_in_,C) basins, during eight-electron reduction. As was previously shown [51], the population of the basins could reflect the strength of interactions between fragments. Thus, the interaction energy between the Re_6_S_8_Cl_2_ cluster fragment and the [bipy_4_]^n^ ligand fragment in different oxidation states (n = 0, 2–, 4–, 6– and 8–) was calculated and compared with the population of the V(Re,N_in_) basins. The linear dependency between the interaction energy (E_int_) of the fragments and the V(Re,N_in_) basin population was obtained (Figure 6) with good accuracy (R^2^ = 0.98). The fitting line crosses the *x* axis at a non-zero value, indicating that a population of ~1.0 e must be present on the V(Re,N_in_) basins to compensate the repulsion between the fragments. A similar result was previously obtained for a heterometallic cubane-type cluster [51]. 

## 4. Conclusions

Octahedral cluster complexes of rhenium with electrochemically inactive ligands are incapable of reversible reduction due to the instability of the cluster core when filling metal-centered antibonding unoccupied orbitals. The ability to reversible reduction associated with the filling of π* orbitals of coordinated 4,4′-bipyridine molecules is the key feature of the *trans*-[{Re_6_S_8_}bipy_4_Cl_2_] cluster complex. The aim of this study was to determine which charge states of the reduced *trans*-[{Re_6_S_8_}bipy_4_Cl_2_]^n–^ are energetically most preferable and how the electronic structure of the cluster changes during reduction. Analysis of the formation energies has shown that the intermediate oxidation states (n = 2–, 4–, 6–) of the cluster are more stable in the open-shell configuration, indicating the paramagnetic nature of the reduced species. Such behavior can be explained by the nature of the four lowest unoccupied orbitals of the neutral cluster. Since the four lowest orbitals are delocalized mainly on bipy ligands in *trans*-positions, their energies are very close to each other; thus, the orbitals are almost degenerate and filled according to Hund’s rule. A notable decrease in the Re–N distances upon reduction indicates the enhancement of the bonding between the cluster core and bipy ligands upon reduction, which was confirmed by EDA. The opposite effect was found for Re–Cl bonds. The reduction of these clusters has practically no effect on the bond lengths inside the {Re_6_S_8_} cluster core, which indicates the absence of its destabilization when electrons are localized on apical redox-active ligands. It was also shown that the interaction energy between the cluster core and bipy ligands correlates linearly with the population of the V(Re, N_in_) ELF basin. The behavior, when the basin population correlates with some bond related properties, is intuitive but does not have much confirmation in the literature [51,52,53]. Finally, since other members of the [Re_6_Q_8_L_4_X_2_] (Q=S or Se; X=Cl or Br; L = 4,4′-bipyridine, 4-phenylpyridine) cluster family have a similar electronic structure to [Re_6_S_8_bipy_4_Cl_2_], tendencies made in the current work can most likely also be applied to these compounds.

## Data Availability

The data that support the findings of this study are available from the corresponding author upon reasonable request.

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
