# Peer review of "Evolution of the Electronic Structure of the trans-[Re6S8bipy4Cl2] Octahedral Rhenium Cluster during Reduction"

_molecules, 2023, doi:10.3390/molecules28093658_

Round 1

Reviewer 1 Report

In the manuscript by Ryzhikov, Gayfulkin, Ulantikov, Arentov, Kozlova and Mironov an analysis of the interaction energy and electronic structure of a rhenium cluster with active ligands during reduction is reported.

To this end, the authors calculate the optimized geometry and conduct frequency calculations of the neutral rhenium cluster and the corresponding results of incorporating two, four, six and eight electrons, using good level-DFT calculations. They find that the most stable electronic structures correspond to the maximum multiplicity possible in each case, according to the Hund rules: the triplet state for charges -2 and -6, and quintuplet state for charge -4. They also conduct an energy decomposition analysis to analyze the interactions between the cluster core, including the chloride unactive ligands, and the active bipyridine ligands. They also report an Electron Localization Function analysis to better describe the changes in bonding during the reductions, and they find that the population of the bonding basins between the Rhenium and the N of the bibypridine ligand bonded to the methal atom increase linearly with the charge of the cluster.

The manuscript describes an interesting work and deserves publication, although some minor points should be corrected by the authors.

  1. On page 4, lines 107-109, the authors explain that in Figure S1 “…four LUMO of bipy’s in cis-positions can form eight MOs…”, but what Figure S1 offers are eight MOs formed from HOMOs of bipy’s.
  2. The Scheme 1 supposedly offers the “changes in charges and spin states”, but the spin states are not indicated in it.
  3. The authors report “average QTAIM charges” in Table S1. A small explanation of how they have obtained these charges would be useful
  4. On line 167 the authors refer to Figure 5, but they mean Figure 6
  5. Some typos and bad writings can be found throughout the manuscript. For instance, the sentence on lines 10-12 in the Abstract is incomplete. Or in line 190 a “they” appear instead of “their”. A thorough revision is needed.

Author Response

Reviewer 1 comments

In the manuscript by Ryzhikov, Gayfulkin, Ulantikov, Arentov, Kozlova and Mironov an analysis of the interaction energy and electronic structure of a rhenium cluster with active ligands during reduction is reported.

To this end, the authors calculate the optimized geometry and conduct frequency calculations of the neutral rhenium cluster and the corresponding results of incorporating two, four, six and eight electrons, using good level-DFT calculations. They find that the most stable electronic structures correspond to the maximum multiplicity possible in each case, according to the Hund rules: the triplet state for charges -2 and -6, and quintuplet state for charge -4. They also conduct an energy decomposition analysis to analyze the interactions between the cluster core, including the chloride unactive ligands, and the active bipyridine ligands. They also report an Electron Localization Function analysis to better describe the changes in bonding during the reductions, and they find that the population of the bonding basins between the Rhenium and the N of the bibypridine ligand bonded to the methal atom increase linearly with the charge of the cluster.

The manuscript describes an interesting work and deserves publication, although some minor points should be corrected by the authors.

  1. On page 4, lines 107-109, the authors explain that in Figure S1 “…four LUMO of bipy’s in cis-positions can form eight MOs…”, but what Figure S1 offers are eight MOs formed from HOMOs of bipy’s.

Response: The incorrect Figure S1 caption has been corrected. The MOs on the figure are composed of LUMO of bipy.

  1. The Scheme 1 supposedly offers the “changes in charges and spin states”, but the spin states are not indicated in it.

Response: This issue was fixed. The spin states have been indicated in the Scheme 1.

  1. The authors report “average QTAIM charges” in Table S1. A small explanation of how they have obtained these charges would be useful

Response: The “Computational details” section has been extended to provide the information about the QTAIM charges calculations.

  1. On line 167 the authors refer to Figure 5, but they mean Figure 6

Response: The reference has been corrected.

  1. Some typos and bad writings can be found throughout the manuscript. For instance, the sentence on lines 10-12 in the Abstract is incomplete. Or in line 190 a “they” appear instead of “their”. A thorough revision is needed.

Response: The manuscript has been revised; typos and bad writings have been corrected

Reviewer 2 Report

The manuscript molecules-2344849 by Ryzhikov et al. focuses on the processes occurring along the redox reactions of transition metal complexes characterized by the presence of redox-active apical ligands. The paper appears well-suited to contribute to the special issue DFT Quantum Chemical Calculation of Metal Clusters”; nevertheless, a few points must be clarified before the scientific contribution may be accepted.

i)               The authors do not say anything about the symmetry assumed to optimize the geometry of the [Re6S8bipy4Cl2]n clusters (n = 0, 2-, 4-, 6-, 8-). Thus, I presume that ADF calculations have been carried out within the C1 symmetry point group. The inspection of Figure 2 reveals the presence of several quasi- degenerate MOs. Usually, the SCF convergence may be tricky in these cases. Did the authors face this problem and eventually how did they overcome it?

ii)              In the Computational Details section, the authors mention the use of the energy decomposition analysis (EDA). Moreover, in the Conclusion section, the authors inform the reader that “Notable decrease of the Re–N distances upon reduction indicates the enhancement of bonding between cluster core and bipy ligands upon reduction, which confirms by the EDA”. Where are the EDA results?  

iii)            In the Computational Details section, the authors mention the use of the electron localization function (ELF). Have the authors tried to compare their ELF results with the bond order indices calculated by the Nalewajski-Mrozek method (available in the ADF-2017 version)?

Minor points

i)               In the abstract, the second sentence (the one starting with “In this works” and ending with “DMS solution”) misses the predicative verb.

ii)              In the introduction (lines 32-34), the authors mention their activity in the last few years without providing any reference to it.

iii)            On p. 2 (line 83), S12h/TZP//S12g/TZP must be modified in S12h/TZ2P//S12g/TZP.

iv)            The acronym QTAIM should be made explicit.    

Academic English is satisfactory; nevertheless, it could be improved.

Author Response

Reviewer 2 comments

The manuscript molecules-2344849 by Ryzhikov et al. focuses on the processes occurring along the redox reactions of transition metal complexes characterized by the presence of redox-active apical ligands. The paper appears well-suited to contribute to the special issue “DFT Quantum Chemical Calculation of Metal Clusters”; nevertheless, a few points must be clarified before the scientific contribution may be accepted.

i) The authors do not say anything about the symmetry assumed to optimize the geometry of the [Re6S8bipy4Cl2]nclusters (n = 0, 2-, 4-, 6-, 8-). Thus, I presume that ADF calculations have been carried out within the C1symmetry point group. The inspection of Figure 2 reveals the presence of several quasi- degenerate MOs. Usually, the SCF convergence may be tricky in these cases. Did the authors face this problem and eventually how did they overcome it?

Response: At first attempt, we tried to use the Ci symmetry (found for the crystal structure of the neutral complex) for all [Re6S8bipy4Cl2]n clusters and then, if optimized structures have imaginary frequencies, C1 symmetry was used for further geometry optimization. In fact, we didn’t have any SCF convergence problems with any charges and spin states. The obtaining of the structures without imaginary frequencies was much more tricky, since the apical bipy ligands are rotationally flexible. We added some discussion of the symmetries and geometry optimization procedure in the “Computational details” section.

ii) In the Computational Details section, the authors mention the use of the energy decomposition analysis (EDA). Moreover, in the Conclusion section, the authors inform the reader that “Notable decrease of the Re–N distances upon reduction indicates the enhancement of bonding between cluster core and bipy ligands upon reduction, which confirms by the EDA”. Where are the EDA results?  

Response: Only the total bonding energy from EDA was used. At the moment we can’t give the full explanation of the EDA results, so in the future some additional calculations of other redox active complexes will be made to understand tendencies that were observed.

iii) In the Computational Details section, the authors mention the use of the electron localization function (ELF). Have the authors tried to compare their ELF results with the bond order indices calculated by the Nalewajski-Mrozek method (available in the ADF-2017 version)?

Response: Since we made direct comparison with bonding energy, Nalewajski-Mrozek method has not been used in this work. However, it can be interesting to compare ELF basins populations with bond order indices for some other complexes where the direct comparison with energy is not possible due to the complexity of the bonds.

Minor points

i) In the abstract, the second sentence (the one starting with “In this works” and ending with “DMS solution”) misses the predicative verb.

Response: The issue has been fixed

ii) In the introduction (lines 32-34), the authors mention their activity in the last few years without providing any reference to it.

Response: The corresponding references have been added.

iii) On p. 2 (line 83), S12h/TZP//S12g/TZP must be modified in S12h/TZ2P//S12g/TZP.

Response: The mistake has been corrected.

iv) The acronym QTAIM should be made explicit.

Response: The definitions of the acronym and QTAIM method itself has been added in the “Computational details” section.

Reviewer 3 Report

1) Did the authors consider the solvent effect in the simulations?

2) which functional was employed?

Author Response

Reviewer 3 comments

1) Did the authors consider the solvent effect in the simulations?

Response: All calculations were preformed with DMSO solvent effect taken into account by Conductor like Screening Model (COSMO).

2) which functional was employed?

Response: The GGA S12g density functional was used for geometry optimization and hybrid S12h density functional was used for consequent single point calculations.